**Diurnal cycle of iodine, bromine and mercury concentrations in Svalbard surface snow**

[1]Andrea Spolaor, [1]Elena Barbaro, [2]David Cappelletti, [1]Clara Turetta, [3]Mauro Mazzola, [4]Fabio Giardi, [5]Mats P. Björkman, [6]Federico Lucchetta, [1]Federico Dallo, [7]Katrine Aspmo Pfaffhuber, [8]Hélène Angot, [9]Aurelien Dommergue, [10]Marion Maturilli, [11]Alfonso Saiz-Lopez, [6,1]Carlo Barbante and [1]Warren RL Cairns

[1]*Institute of Polar Science, IDPA-CNR, Campus Scientifico Via Torino 155, 30172 Mestre, Venice, Italy.*

[2]*Dipartimento di Chimica, Biologia e Biotecnologie, Università degli Studi di Perugia I-06123 Perugia, Italy*

[3]*National Research Council, Institute of Atmospheric Sciences and Climate, CNR-ISAC, Via P. Gobetti 101, Bologna, Italy*

[4]*Chemistry Department – Analytical Chemistry, Scientific Pole, University of Florence, Via della Lastruccia 3, I-50019 Sesto Fiorentino (Florence) Italy.*

[5]*University of Gothenburg, Department of Earth Sciences, Box 460, 40530 Göteborg, Sweden*

[6]*Ca' Foscari University of Venice, Department of Environmental Sciences, Informatics and Statistics, Santa Marta – Dorsoduro 2137, 30123 Venice, Italy.*

[7]*NILU - Norwegian Institute for Air Research, Kjeller, Norway*

[8]*Institute of Arctic and Alpine Research (INSTAAR), University of Colorado, Boulder, USA*

[9]*Institut des Géosciences de l'Environnement, Univ. Grenoble Alpes, CNRS, IRD, Grenoble INP, 38000 Grenoble, France.*

[10]Alfred Wegener Institute, Helmholtz Centre for Polar and Marine Research, Potsdam, Germany

[11]*Department of Atmospheric Chemistry and Climate, Institute of Physical Chemistry Rocasolano, CSIC, Madrid, Spain*

**Abstract**

Sunlit snow is highly photochemically active and plays a key role in the exchange of gas-phase species between the cryosphere to the atmosphere. Here, we investigate the behaviour of two selected species in surface snow: mercury (Hg) and iodine (I). Hg can deposit year-round and accumulate in the snowpack. However, photo-induced re-emission of gas-phase Hg from the surface has been widely reported. Iodine is active in atmospheric new particle formation, especially in the marine boundary layer, and in the destruction of atmospheric ozone. It can also undergo photochemical re-emission. Although previous studies indicate possible post-depositional processes, little is known about the diurnal behaviour of these two species and their interaction in surface snow. The mechanisms are still poorly constrained, and no field experiments have been performed in different seasons to investigate the magnitude of re-emission processes Three sampling campaigns conducted at an hourly resolution for 3-days each were carried out near Ny-Ålesund (Svalbard) to study the behaviour of mercury and iodine in surface snow under different sunlight and environmental conditions (24h-darkness, 24h-sunlight and day/night cycles). Our results indicate a different behaviour of mercury and iodine in surface snow during the different campaign. The day/night experiments demonstrate the existence of a diurnal cycle in surface snow for Hg and iodine, indicating that these species are indeed influenced by the daily solar radiation cycle. Differently bromine did not show any diurnal cycle. The diurnal cycle also disappeared for Hg and iodine during the 24h-sunlight period and during 24h-darkness experiments supporting the idea of the occurrence (absence) of a continuous recycling/exchange at the snow-air interface. These results demonstrate that this surface snow recycling is seasonally dependent, through sunlight. They also highlight the non-negligible role that snowpack emissions have on ambient air concentrations and potentially on iodine-induced atmospheric nucleation processes.

## 1. Introduction

Polar Regions are being increasingly studied for their important roles in global climate and atmospheric chemical cycles. Multiple studies have improved our understanding of atmospheric processes in polar regions, ranging from new particle formation processes (Dall´Osto et al., 2017; Sipilä et al., 2016), ozone destruction processes (Saiz-Lopez et al., 2007; Simpson et al., 2007), the role of halogens in polar atmospheric processes (Saiz-Lopez and von Glasow, 2012; Spolaor et al., 2013a), the mercury cycle (Angot et al., 2016a; Aspmo et al., 2005; Brooks et al., 2006; Dommergue et al., 2003a; Durnford and Dastoor, 2011; Skov et al., 2006) to atmospheric transport and deposition of natural and anthropogenic compounds (Moroni et al., 2015; Moroni et al., 2017; Udisti et al., 2016; Zangrando et al., 2013). The Polar Regions are characterized by periods with 24 h of continuous solar radiation (April to September in the Arctic), periods when the night and day cycle is present (February to March and September to October in the Arctic) and periods of continuous darkness (November to January in the Arctic), the so-called polar night. The different periods have completely different environmental conditions depending on the incoming solar radiation, with variables such as sea ice presence or biological activity being radically altered by sunlight. One important aspect is snow cover. Annual snow is present, on average, for almost nine months of the year and represents an important environmental component of Polar Regions. In Svalbard, the snow starts to accumulate in October and remains until the end of May when the melting season begins (Førland et al., 2011). However, with Arctic temperatures rising (Maturilli et al., 2013), the length of the snow cover has diminished (Brage B. Hansen et al., 2014), with direct consequences on the environment of the Svalbard archipelago, such as glacier mass loss, permafrost thawing, disturbances of the local fauna etc. (Karner et al., 2013; Kohler and Aanes, 2004; Kohler et al., 2007; Westermann et al., 2011). The annual snow layer is an extremely dynamic portion of the cryosphere, and can be defined as the snow accumulated and present on the ground during the whole year (Spolaor et al., 2016a). The characteristics of the annual snow strata are strongly dependent on climate conditions and may influence food access for animals that rely on food sources below the snow (Kohler and Aanes, 2004). From a chemical point of view, snow is a sink for an impressive number of chemical compounds (natural and anthropogenic) and elements (Björkman et al., 2013; Gabrieli et al., 2011; Vecchiato et al., 2018). Specific compounds and elements accumulate during the winter can undergo photo-activation and can be re-emitted into the atmosphere (Angot et al., 2016c; Spolaor et al., 2014), while taking part in numerous geochemical and biological cycles (Björkman et al., 2014) during spring and summer. Mercury (Hg) and iodine (I) are two elements that can be photo activated and released from the snow pack. Mercury is a heavy metal with a known toxicity present in the environment in several different chemical forms. It is reactive in the environment and undergoes photochemical reactions that change its speciation and chemical behaviour (Dommergue et al., 2010; Durnford and Dastoor, 2011; Saiz-Lopez et al.,

2018; Steffen et al., 2002). Mercury in its oxidized form can be deposited onto the snowpack,
increasing Hg concentrations in the upper snow strata (Obrist et al., 2017). Once present in the
snowpack, Hg is very labile, and it can be reduced back to elemental Hg (Hg(0)) and undergo
dynamic exchange with the atmosphere (Song et al., 2018; Spolaor et al., 2018; Steffen et al., 2002).
The role of the snowpack is crucial in the mercury cycle in Polar Regions since it acts as both a sink
(deposition, accumulation) and a source (re-emission). Several studies have already been carried out
in the polar regions with the aim of determining the extent of mercury recycling between the
surface snow and the lower atmosphere (Angot et al., 2016c; Brooks et al., 2008; Brooks et al.,
2006; Dommergue et al., 2012; Douglas et al., 2008; Han et al., 2014; Obrist et al., 2017; Wang et
al., 2016). It has been shown that surface Arctic snow could lose up to 90 % of its total Hg content
within 48 hours (Poulain et al., 2004). Similar, re-emission/loss rates of Hg from snow surface (35–
50 %) and drifting snow (65–75 %) over 10.5 h have been suggested in chamber experiments
(Sherman et al., 2010) while, in a study performed on the Antarctic Plateau, Spolaor et al. 2018
suggest a loss of 90% of mercury in the upper snow layer within a few hours. High gaseous
elemental mercury (GEM) emission from the snow pack has also been determined at Station North
(Greenland) where the emission flux can rise up to 190 ng m$^{-2}$ min$^{-1}$ (Kamp et al., 2018). Similar to
mercury, iodine can undergo photochemical activation in surface snow resulting in its presence in
the surrounding atmosphere (Frieb et al., 2010; Spolaor et al., 2014). Several studies aimed at
understanding the behaviour of iodine in the Arctic region, from a paleo perspective using ice core
archives (Cuevas et al., 2018; Spolaor et al., 2016b), and field (atmospheric and snow) experiments
(Frieb et al., 2010; Gilfedder et al., 2007). The role of iodine in new particle formation as well in
ozone destruction is currently under investigation (Allan et al., 2015; Saiz-Lopez et al., 2012; Saiz-
Lopez, 2006; Sipilä et al., 2016) since it could have a direct effect on the radiative budget of polar
areas. Up to now, it was believed that iodine was mainly associated with biological emissions,
however, recent studies have underlined the increase in ocean inorganic emissions (tripled since
1950) connected with the increase in anthropogenic ozone via reactions over the ocean surface
(Cuevas et al., 2018). Like mercury, iodine could be released from surface snow and directly
participate in specific processes within the marine boundary layer, particularly in new particle
formation. Little information exists on the behaviour of mercury and iodine in surface snow during
different seasons. Laboratory experiments have been carried out to understand light-induced
processes regarding Hg and iodine (Durnford and Dastoor, 2011; Saiz-Lopez et al., 2012; Spolaor et
al., 2013b). However few experiments have been carried out in the field, with the specific aim of
understanding the diurnal dynamics of these elements in surface snow (Dommergue et al., 2003b;
Ferrari et al., 2005; Spolaor et al., 2018). The unique high-temporal resolution experiments
presented, aim to improve our understanding of the behaviour of these elements in the upper snow
layers (0-3 cm) under different light and atmospheric conditions to investigate their short-term
(diurnal) variation.

**2. Methods**
The experiments were conducted in the vicinity of Ny-Ålesund, in the snowfield behind the
Gruvebadet aerosol site (Figure 1). This area has a homogeneously flat surface without specific
elevation changes or obstacles that might interfere with snow deposition or wind-blown
redistribution. This area is approximatively 1 km from the coast line of the Kongsfjorden and about
400 m from the Zeppelin mountain. The "Gruvebadet" snow field is located to the south of Ny-
Alesund at an elevation of the 80 m a.s.l. (Figure 1). The "Gruvebadet" snow field is located to the
south of Ny-Ålesund, while the prevailing wind are mainly from east and south-east, minimizing
possible influences from station activities.


**2.1 Sampling period and strategy**
Surface snow samples were collected in the vicinity of Ny-Ålesund, specifically in the snow field
behind the "Gruvebadet" Aerosol Laboratory (Figure 1). Three experiments were conducted, two in
spring (2015 and 2016) and one in winter (2017). In 2015, we performed the first surface
experiment (hereafter called the "2015 experiment") between the 28th of April and the 1st of May.
This period was characterized by 24 hours of sunlight (incoming solar radiation ranged from a
minimum of 25 Wm$^{-2}$ to a maximum of 456 W/m$^2$). In 2016, a second experiment (hereafter called
the "2016 experiment") was carried out between the 6th and the 9th of April when the night and day
cycle was still present at Ny-Ålesund (incoming solar radiation between 0 and 227 Wm$^{-2}$). The last
experiment was conducted during the polar night, between the 24th and the 29th of January 2017
(hereafter called the "2017 experiment") with the complete absence of incoming solar radiation.
The 2017 experiment was conducted during the second half of January when full snow cover is
guaranteed (López-Moreno et al., 2016). In December, snow cover in the Spitsbergen area is not
homogeneously distributed. The ground could still be partially exposed, meaning that locally
generated windblown dust could affect the trace element concentrations in the snow surface. The
spring period selected for the 2016 experiment had two main characteristics: a well-defined night
and day cycle without a long sunset, avoiding possible incoming solar radiation by diffraction
processes over the horizon. There was also the possibility to observe atmospheric mercury depletion
events (AMDE) connected with bromine explosion events (Lu et al., 2001; Moore et al., 2014;
Schroeder and Munthe, 1998). Unfortunately, these events were not observed as the northern coast
of Svalbard was virtually ice free by the time we started sampling. The 2015 experiment was
scheduled to end at the beginning of May, when we have a full 24 h of sunlight reaching the snow
surface, but temperatures are still below freezing, avoiding/minimizing the confounding effects of

snow pack melt or collapse on surface photochemical processes and gaseous mercury transport in the interstitial air. The meteorological conditions throughout all the experiments are within the expected local conditions for the time of year.

To determine the diurnal variation, and the rates of the expected changes in iodine and mercury concentrations, a high temporal resolution (hourly) sampling strategy was adopted. An area of approximately 2 m x 2 m was delimited for surface snow sampling, and all samples were collected inside this delimited area. At the beginning of the experiment, six samples were collected to evaluate the spatial variability of mercury, iodine, bromine (bromine is limited to the "2016 experiment") and sodium in surface snow within the delimited snowfield. Afterwards, surface snow (the first 3 cm) was sampled with an hourly resolution for three consecutive days. The upper 3 cm were chosen as this is the snow layer that is most influenced by the surrounding atmospheric conditions, and, in case of snowfall, by deposition (Spolaor et. al. 2018). This choice also minimizes the effect of different physical snow conditions (density and crystal shape and size). Although re-emission of mercury and iodine from lower snow strata could influence the gaseous concentrations in the snow interstitial air (Faïn et al., 2007) it is much less likely to have a direct effect on snow concentrations due to its poor solubility in water. During snow sampling, the temperature of surface snow was also measured. To minimize spatial variability, samples were collected following a straight line leaving about 5 cm between each of the sampling points. After collection, the snow samples were stored at -20°C in dark conditions and transported to the Venice IDPA-CNR laboratories. The samples were never melted or exposed to direct sunlight until analysis.

## 2.2 Meteorological measurements

Meteorological and radiation conditions were monitored at the Amundsen-Nobile Climate Change Tower (Mazzola et al., 2016), located about 500 m west of the sampling site and from the AWIPEV observatory (Maturilli et al., 2013), located about 800 m north of the sampling site. No meteorological measurements are present in the sampling area. Temperature, relative humidity, were measured at 2 m above ground level and were considered as representative of the atmosphere just above the snow surface while wind speed and direction at 10 m above ground. Incoming solar radiation was measured at the top of the CCT tower (33 m), this value was not influenced by reflections from the structure. One-minute data were used to obtain hourly averages. Snow accumulation data were obtained by measuring the high of 4 plastic poles located at the extremities of the snow sampling field. Precipitation data were recorded in Ny-Ålesund by the Norwegian Meteorological Institute (station n. 99910) and downloaded through the eKlima database (eklima.no).

## 2.3 Snow Mercury analysis

Total Hg concentrations in surface snow samples was determined using a Thermo Element Inductively Coupled Plasma Sector Field Mass Spectrometry (ICP-SFMS Element XR, Thermo-Fisher, Bremen, Germany) in low resolution scanning mode using $^{202}$Hg as the analytical mercury mass with 10 replicates per sample measurement. The instrument was calibrated using standards prepared from a mono-elemental Hg solution (TraceCert®, purity grade, Sigma-Aldrich, MO, USA). Hg calibration standards were re-analysed every 10 samples as a quality control check. The percent relative standard deviation (n=10) ranged from 0.5 % at 500 pgg$^{-1}$ to 10 % at 1 pgg$^{-1}$ and amounted to 2.6 % on average. Considering the high volatility and instability of Hg in solution, the samples were acidified at 2 % v/v with ultrapure hydrochloric acid before they were melted and analyzed. Each sample was weighed and the exact amount of HCl was added to reach a final concentration of 2 % (Planchon et al., 2004; Spolaor et al., 2018).

## 2.4 Snow iodine, sodium and bromine analysis

Halogens (I and Br) and sodium analyses were conducted on non-acidified samples. Total sodium (Na), iodine (I) and bromine (Br) concentrations were determined by ICP-SFMS (Spolaor et al., 2016c). Each analytical run started and ended with an ultra-pure water (UPW) cleaning session of 3 min to ensure a stable background level throughout the analysis. The external standards that were used to calibrate the analytes were prepared by diluting a 1000 ppm stock IC (ion chromatography) standard solution (TraceCERT® purity grade, Sigma-Aldrich, MO, USA). The standard concentrations ranged between 10 and 4000 ng g$^{-1}$ for sodium, 0.01 and 1 ngg$^{-1}$ for iodine and between 0.5 and 20 ng g$^{-1}$ for bromine. The residual standard deviation (RSD) was low for all analytes, the halogens ranged between 1–2% and 2–5% for Br and I, respectively, and the RSD was 3-4 % for sodium.

## 2.5 Atmospheric mercury measurements

Atmospheric mercury concentrations were obtained from the Zeppelin Observatory located at 474 m a.s.l, less than 1 km away from the sampling site (Figure 1). Gaseous elemental mercury (GEM) was monitored using a Tekran 2537 Hg vapor analyzer as described by Aspmo et al., 2005 and as summarized here: ambient air was sampled at 1.5 l min$^{-1}$ through a Teflon filter via a heated sampling line. A soda-lime trap was mounted in-line before the instrument filter. Hg in the air is pre-concentrated for 5 minutes by amalgamation on two parallel gold cartridges, which alternate between collection and thermal desorption, followed by AFS (atomic fluorescence spectrometric) detection. The instrument was auto-calibrated every 25 hours using an internal Hg permeation source, whose accuracy was verified during routine site audits that include manual injections of Hg from an external source (Aspmo et al., 2005). The measurements at Zeppelin were the only GEM measurements available in the Ny-Alesund area. Although GEM measurements at the snow

sampling site would have been more reliable in determining possible interactions between snow and atmospheric mercury, it was not possible to set up an instrument at the site. We assume that the snow reactions occurring at the sampling site at 40 m a.s.l. are of the same order of magnitude as those occurring in the snow layers surrounding Zeppelin station.

## 3. Results

The 2015 and 2016 experiments were characterized by similar atmospheric conditions (except for the incoming solar radiation) while during the 2017 experiment a storm approached Ny-Ålesund during the first 12 hours of the experiment with strong winds lasting for the first 24 hours of the experiment. During the 2015 experiment under full day conditions, the average air temperature ranged between -10°C and -6°C and the surface snow temperature range between -13°C and -5°C, showing a diurnal variability connected with changes in the incoming solar radiation (Figure 2). Incoming solar radiation ranged from a minimum of 25 $Wm^{-2}$ to a maximum of 450 $Wm^{-2}$. Wind speed was almost constant and remained below 3 $ms^{-1}$ during most of the experiment, except for a few hours at the beginning when it exceeded 3 $ms^{-1}$. Snowfall (1 cm net accumulation on the ground) occurred on the 30[th] of April between 3 am and 11 am (Figure 2, pink rectangle). A snow event, causing a net accumulation of 1 cm of snow, also occurred during the 2016 experiment (Figure 3, pink rectangular) when day and night periods were present. The snow event occurred on the 9[th] of April between 10 am and 3 pm. During the 2016 experiment, conducted between 6[th] and the 9[th] of April, the snow temperature was not registered due a technical problem with the temperature probe installed in the snow. Air temperature ranged between -7 and -3°C and solar radiation between zero at night time to a maximum of 227 $W/m^2$. As for the first experiment, wind speed was below 3 $ms^{-1}$, minimizing the effect of blowing snow. Wind direction was almost constant and prevailing from east. The GEM and the surface snow mercury datasets were de-trended to emphasize the diurnal variation and remove the decreasing trend present in both datasets. The de-trended series were obtained calculating the linear regression line for both series and subtracting this value from the data. Figure 3 (middle panel) reports the de-trended mercury dataset while the Figure 4 shows the raw data and the methods used to remove the trend. The 2017 winter experiment (Figure 5) was characterized by a snowstorm that occurred on the 24[th] of January (10 hours after the experiment began, pink rectangle). Differently to the previous experiments, the wind speed averaged 9 $ms^{-1}$ during the storm, with a maximum speed of 16 $ms^{-1}$ (Figure 5, orange line). Strong winds can redistribute surface snow and significantly change chemical concentrations. For these reasons, the winter experiment began on the 24[th] of January and ended on the 29[th] of January for 5 days in total, compared the 3 days adopted for the 2015 and 2016 experiment. The length of the experiment was extended of 2 days to minimize the impact of the strong wind and snowfall that occurred at the beginning of the experiment. Air temperatures ranged from between -17°C and -3°C,

while snow temperatures ranged between -25°C and -10°C (Figure 5, upper panel). One important
issue that could confound the results obtained by surface sampling is spatial variability. Spatial
variability was tested during the three experiments and specifically for the four elements
investigated. Six surface snow aliquots were collected at the beginning of each experiment within
the delimited area at the same time. The results obtained show that for sodium, bromine and
mercury, spatial variability can explain 10% of the variability whilst for iodine the variability was
of the order of 5%. Concentrations detected during the three experiments show different
background levels (Table 1) for total iodine, sodium, mercury and gaseous elemental mercury (Br
was measured only during the 2016 experiment). For sodium, the highest concentration was
detected during the 2015 (full day) experiment where concentrations in surface snow averaged 3500
$ngg^{-1}$. The lowest sodium concentrations were determined during the winter period with
concentrations of around 1500 $ngg^{-1}$. For iodine the trend was the opposite, with highest
concentrations in winter (0.38 $ngg^{-1}$) and the lowest during the 24h sunlight period (0.15 $ngg^{-1}$). For
total mercury, the minimum concentration was found in early spring (0.007 $ngg^{-1}$, 2016 experiment)
while the highest values were detected during 2015 (full light) and 2017 winter experiment (on
average 0.010 $ngg^{-1}$ for 2015 and 0.009 $ngg^{-1}$ for 2017). Gaseous elemental mercury during the
experiments had the highest concentration during springtime, when 24 h incoming solar radiation is
present (1.45 $ngm^{-3}$) while the lowest value has been detected during the polar night (1.28 $ngm^{-3}$).
The average concentration during the experiment is only representative for specific periods in the
experiment and should not be considered as a reference concentration for a specific season. The
experimental periods were chosen to reduce the possibility of snowfall deposition during the
experiment and to avoid periods with strong wind and subsequent windblown snow transport (the
main reason why the winter experiment was lengthened to 5 days). This was all done to minimize
the effects of meteorological parameters on our results and make the experiments more comparable.
We cannot exclude that the behaviour that we found for iodine, mercury and bromine could be
significantly different during the specific season/periods (such as for example during an AMDE) or
when meteorological conditions such as snow deposition frequency and amount, wind strength and
cloud coverage were different. Some indications emerged, especially for iodine, which showed the
highest concentrations during the polar night in the absence of solar radiation. Considering iodine
(inorganic and organic) is mainly emitted by oceanic processes, iodine concentrations were
normalized to sodium concentrations to obtain an iodine enrichment ($I_{enr}$) compared to the bulk
seawater abundance. This is defined as $I_{enr}=I_{snow}$ x ($Na_{snow}$* $[I/Na]_{sw}$, where I/Na = 0.00000596,
(Millero et al., 2008) where *"sw"* = measured sea water abundance. In the 2015 experiment (24h
sunlight), iodine had an average enrichment value of 5, a value that exponentially increased (up to
190) during snowfall (Figure 2), so if we consider the snowfall period the mean value increases to
10. The 2016 experiment (day/night) was characterized by a diurnal cycle for both mercury and
iodine (and $I_{enr}$) and by an average $I_{enr}$ value of 11, with the lowest value during day time and higher
values detected during the night periods (Figure 3). As for the 2015 experiment, the experiment
conducted in 2016 was characterized by a snowfall event that significantly influenced the surface
iodine concentration and its enrichment factor. During the 2016 experiment, snowfall caused the $I_{enr}$
to increase up to 300. The rapid increase in iodine and its enrichment factor during snowfall was
followed by a rapid decrease to the pre-snowfall (seasonal background value) concentration during
the 2015 experiment (Figure 2), whilst in the 2016 experiment the increased concentration and
enrichment caused by the snow fall was most likely masked by the night time deposition (Figure 3).
Similar behaviour was measured for total mercury in surface snow samples, with an increase in
concentration during snowfall followed by a rapid decrease in both experiments (Figure 2 and 3).
The winter experiment is characterized by the highest iodine enrichment values (47 on average) and,
similar the previous experimental results, the experiment was characterized by snowfall and strong
winds during the first 24h. During the storm period in the winter experiment we detected an
increase in iodine concentrations (and $I_{enr}$ up to 100), however the difference in iodine enrichment
between the snowfall periods and rest of samples collect was not statistically significant. The
average elemental concentrations for each experiment are reported in Table 1.

## 4. Discussion

The behaviour of mercury and iodine in surface snow depends on the season and the amount of
incoming solar radiation (Figure 2, Figure 3, Figure 5). Atmospheric mercury depletion events
(AMDE) can occur during the springtime causing large-scale deposition of mercury to the snow
pack concurrently with ozone photochemistry and oxidation reactions involving bromine. During
our spring experiments we have not observed any rapid decreases in GEM or increases in mercury
concentrations in the surface snow. This indicates that no AMDE occurred during the sampling
periods and that, especially for bromine, the main depositional source was from sea spray given the
distance from the coast line (< 1km) and the positive correlation with Na (Table 2). This is inline
with the findings of (Angot et al., 2016a), who reported that AMDEs occur much less frequently at
Zeppelin station than they do at Alert or Station North in Greenland.
During wintertime (Figure 5), iodine behaves similarly to sodium. Sodium does not undergo
photochemical processes in the snow and is often used to evaluate/correct for marine sea spray
emission/deposition (Spolaor et al., 2014). During winter, iodine has higher concentrations and
enrichment factors (compared to its sea water abundance based on the I/Na mass ratio). These
higher values in surface snow could be due to the absence of photochemical activation by solar
radiation. In the absence of photochemistry and with limited biological production in winter
(Ardyna et al., 2013), we expect to find enrichment values close to the seawater abundance.

However, during the 2017 experiment, iodine had higher than expected enrichment values suggesting that an extra source(s), in addition to sea spray emission may exist and that it might be dominant during winter. Saiz-Lopez et al. 2016 suggests that nighttime radical activation can occur. They indicate that the reaction of HOI with $NO_3$, to yield $IO + HNO_3$, is possible under winter tropospheric conditions (Saiz-Lopez et al., 2016). The inclusion of this reaction, along with that of $I_2 + NO_3$, has a number of significant implications, such as the night time activation of iodine radical chemistry that can cause an enhanced night time oceanic emissions of HOI and $I_2$ (Saiz-Lopez et al., 2016). Although typical $NO_x$ levels are low in the Arctic, the reaction with $NO3$ could be relevant close to Arctic cities and under episodes of anthropogenic long range transport of pollution to the Arctic. Sea spray aerosol droplets could absorb gas phase iodine emissions from the ocean surface (as suggested by the high correlation between total iodine and sodium, Figure 5 and Table 2) and deposited on the surface snow causing the high iodine surface snow enrichment. This process, together with the absence of photoactivation that causes iodine loss from the snow surface, could explain the high level of iodine during the polar night.

In parallel to iodine, our experiments have focused on the rapid changes in mercury concentrations that could occur in surface snow during the polar night. This is because without these temporal resolution measurements, it is extremely difficult to determine which reactions might be occurring. During the first 24 hours of the winter experiment (2017) we had strong winds remodelling the snow surface. Variations in surface mercury concentrations detected within the first 24 hours may in part have been due to snowfall and physical artefacts caused by windblown snow redistribution. After the storm, total mercury concentrations in surface snow tended to stabilize until the end of the experiment. It should be noted that some oscillations in surface snow mercury concentrations and the ambient air above have been detected. Mercury in the snow rapidly decreased from 00:00 on the 24[th] until noon on the same day and was associated with an increase in the atmospheric mercury concentration (Figure 5). After this sharp increase, the GEM concentration decreased rapidly while the surface snow mercury increased. These two rapid events occurred within about 24 hours, supporting the idea of a connection and interchange between GEM and the mercury present in snow surface, even during the night-time. Night-time mercury reactions have been thought to occur. Angot et. al. 2016b suggested that mercury deposition onto the snow surface in the dark could be due to several mechanisms, including gas-phase oxidation, heterogeneous reactions, or dry deposition of Hg(0) (Angot et al., 2016b; Song et al., 2018). This hypothesis however is based on results obtained at Dome C on the Antarctic Plateau over the entire winter season, conditions very different to those in Svalbard. The average mercury surface snow concentrations detected during the winter experiment are comparable to those during the 2015 experiment (Figure 2 and Table 1), this might be due, as for iodine, to the lack of light induced snow re-emission, but might also be caused by different background concentrations independent of any seasonal effect.

The most interesting experiment is the one conducted during early April in 2016 when a day and
night cycle was still present (Figure 3). During this experiment, mercury and iodine, show a similar
pattern with a distinct diurnal cycle in surface snow ($I_{snow}$ vs $Hg_{snow}$ R = 0.57 p-value < 0.01). For
both elements, the highest concentrations were detected during the night and the lowest during the
day. Iodine has been demonstrated to be active in the upper snow layer. Previous laboratory and
outdoor experiments have demonstrated two photo-induced mechanisms for the release of inorganic
iodine from the snowpack to the atmosphere: i) photooxidation of iodide in ice with the resulting
production of tri-iodide ($I_3^-$) and evaporable molecular iodine ($I_2$) (Kim et al., 2016), and ii) the
emission of an iodine photofragments following the heterogenous photoreduction of iodate in ice
(Gálvez et al., 2016). These experimental studies have shown that the release of iodine from the
snow/ice to the atmosphere depends on solar radiation. Indeed, (Raso et al., 2017) recently
measured $I_2$ in the Arctic atmosphere under natural sunlight conditions with results that are in
agreement with the supposed photochemical production mechanisms.
Kim et al., 2016, showed that the iodide photooxidation to $I_3^-$ strongly depended on irradiation time
following the UV-Visible absorption spectrum of iodide in ice. This would explain the observations
of reduced iodine concentrations in ice during the sunlit parts of the day. Although we do not have
observations of atmospheric iodine, it is very likely that snow re-emission during the day leads to a
peak in reactive gas phase iodine in the overlying polar boundary layer at low solar zenith angles.
The emitted gas phase iodine would then readily form reservoir species (HOI, $IONO_2$, HI) (Saiz-
Lopez et al., 2014) that once photochemistry ceases could deposit and accumulate in the  snow/ice
until the following sunrise, when re-emission starts again.
Active mercury recycling from the snow pack has already been suggested/observed by several
authors (Dommergue et al., 2012; Durnford and Dastoor, 2011; Song et al., 2018; Steffen et al.,
2008). Mercury in its oxidized forms can be deposited onto the snowpack, increasing total Hg
concentrations in the upper snow strata. Once present in the snowpack, Hg is very labile, it can be
reduced back to Hg(0) and can undergo dynamic exchange with the atmosphere above (Steffen et
al., 2002). Atmospheric mercury can undergo wet or dry deposition to the snow pack, either as
gaseous elemental (GEM) or oxidised mercury (GOM), and can be reemitted as GEM (Brooks et al.
2006). Photochemical reactions are important in altering the speciation of Hg in the snowpack and
depend on environmental properties and snowpack chemistry. Spolaor et al. 2018 shows that total
Hg concentrations in the surface snow in the inner Antarctic Plateau do not exhibit a clear diurnal
cycle as has been determined for gaseous elemental mercury (Angot et al., 2016c; Dommergue et
al., 2012). However Hg in surface snow shows the highest values during the insolation minima,
suggesting that its concentration in the snow might be influenced by daily differences in incoming
solar radiation. The experiment at Dome C (Spolaor et al. 2018) was carried out under full polar day
conditions with incoming solar radiation reaching the snow surface for the entire period of the

experiment. The experiment conducted at Ny-Ålesund between the 6[th] to the 9[th] of April 2016 was characterized by a night and day cycle. Similar to iodine, a clear diurnal cycle has been detected for atmospheric and surface snow mercury. Snow mercury shows the highest concentrations during the night, with a minimum during the daytime (night periods are highlighted in Figure 3 by the grey rectangular). Contrary to this, the GEM shows a minimum during the night-time and a maximum during the daytime. This anti-phase behaviour (Figure 3 and Table 2) suggests that under day light conditions, mercury in the surface snow can be reduced and released by photochemical processes from the snow surface, resulting in increases in atmospheric concentrations. This is not the only mechanism that can lead to increases in atmospheric concentrations. Changes in the atmospheric mixing layer height may lead to apparent concentration changes of atmospheric species, even if total amounts in the boundary layer remain constant. In the Ny-Ålesund area it is difficult to estimate the height of the boundary layer due to effects induced by winds and by the orography of the Brøgger Peninsula. However, during the experiments the stable meteorological conditions suggested that the atmospheric mixing height was quite stable, minimizing any influence of the boundary layer on GEM concentrations.

During the night, mercury can be oxidized to Hg(II) and re-deposited onto the snow surface. In addition to this diurnal oscillation during the experiment, if we exclude the snowfall that caused a re-enrichment of surface snow for both elements, we detected a decreasing trend for mercury in snow as well as in the atmosphere (Figure 4 and Table 2), from the beginning to the end of the experiment. This decreasing trend may be ascribed to re-emission during the daytime and an incomplete deposition during the night due to possible dilution/removal processes caused by the surrounding atmosphere, with air mass movements as well for mixing within the upper atmospheric strata. This suggested atmospheric removal could explain the positive correlation between GEM and snow surface Hg seen in Table 2 that is masking the antiphase caused by the diurnal daylight cycle. When the two series are de-trended by removing the overall decreasing trend, (by considering 6-hour average values) the correlation between atmospheric and snow mercury becomes significantly negative (Figure 4 upper panel and Table 2).

At the end of the 2016 experiment, a snowfall event occurred (Figure 3 pink rectangular). The net effect of the snowfall was to increase the mercury concentration in the upper snow surface. Precipitation events seem to be associated with elevated total Hg concentrations in surface snow samples (Figure 2 and Figure 3). Angot et al., 2016c have suggested that the presence of ice crystals could enhance the dry deposition of Hg(II). Indeed, due to an elevated specific surface area, the mercury-capture efficiency of ice crystals is high (Douglas et al., 2008). Although there is a deposition of mercury to surface snow, atmospheric mercury did not show a decrease in concomitance with the snowfall but continued to show the usual diurnal pattern. In Antarctica, it has been demonstrated that snow and atmospheric mercury concentrations are related but it should

be taken into consideration that the boundary layer can be confined to the first 30 meters above the
snow surface (Angot et al., 2016a). After the snow fall the mercury surface snow concentration
decreased from 45 to 8 pgg$^{-1}$ with a net loss of 37 pgg$^{-1}$ of total mercury in 1 hour. Assuming all
snow mercury lost is lost as GEM, considering a sampling depth of 3 cm for an area of 1 m$^2$ and
considering an average snow density of 0.3 g cm$^{-3}$, the emission rate would be 5.5 ng m$^{-2}$ h$^{-1}$, a
similar order of magnitude to that determined by Kamp et al. 2017. It must be noted that Kamp et al.
2017 measured the total emission flux while we focussed on the upper snow pack layer, emissions
from the lower/deeper strata are not considered that might contribute to the total emission from the
snow pack. The mercury released from the snow after snowfall may not be enough to impact the
GEM due to dilution effects. Is also possible that the Zeppelin station is located often above the
marine boundary layer and the mercury released from the snow is confined and is not able to
influence the mercury concentration in the free troposphere. Zeppelin station is at a higher elevation
(approximately 400 meters above the sampling site) compared to the snow-sampling site, but is the
only site giving hourly mercury atmospheric measurements in the area. Although the two sites may
not be directly connected (Aspmo et al., 2005), we assume that the snow mercury and iodine release
mechanisms that occur in the snow at our sampling site are also occurring in the snow surrounding
the Zeppelin station at more or less the same rates. Consequently, GEM atmospheric concentrations
and the diurnal cycle should be representative of the variations in the atmospheric cycle above the
surrounding sampled snow field.
Surface snow iodine concentrations, similarly to mercury, are enhanced during liquid or solid
deposition. Several studies have demonstrated that rain, snow and aerosol are enriched in soluble
organic iodine as well as inorganic iodine (iodide and iodate) (Baker, 2005; Saiz-Lopez and von
Glasow, 2012). Uptake of iodine species by cloud droplets and snowflakes followed by wet
deposition or snowfall are major atmospheric iodine removal processes, which would enhance the
concentration of iodine in the snow/ice. It is interesting to note that after the snowfall events, the
enhanced concentrations in surface snow rapidly decrease. This phenomenon is more evident during
the 2015 experiment (Figure 2) when 24 hours of solar irradiation occur. In the 2016 experiment
after the snowfall, the iodine decrease is probably masked by nocturnal deposition. Bromine was
also measured during the 2016 experiment (Figure 6) to understand if, as for iodine and mercury, it
can undergo surface recycling re-emission processes as suggest by previous studies (Simpson et al.,
2007). Bromine shows a correlation of r = 0.85 with sodium and, the Br$_{enr}$ factor (calculated as
Br$_{enr}$=Br$_{snow}$ x (Na$_{snow}$* [Br/Na]$_{sw}$, where Br/Na$_{sw}$ = 0.006), does not show a diurnal cycle as for
iodine (and its enrichment factor) and mercury. As has already been proposed, bromine after
deposition is probably preserved in surface snow (Spolaor et al. 2014). During snowfall, both
sodium and bromine decrease, most likely due to the dilution effect caused by new snowfall. It is
likely that the main sodium and bromine deposition occurred by sea spray deposition caused by
wave breaking (no sea ice was present in the fjord in front of Ny-Alesund during the experiments so
the bromine explosion over sea ice did not occur). Windblown snow and, eventual snowfall, can
affect the deposition of what is present in the atmosphere and dilute the concentrations in surface
snow. However, it should be noted that although Br and Na surface snow concentrations decrease
during snowfall, the Br enrichment factor increased, suggesting that snowfall is able to scavenge
gas phase bromine present in the atmosphere in addition to the aerosol phase and deposit it onto the
snow surface.
The experiment conducted in 2015 was characterized by full light conditions (Figure 2) similar to
those encountered in Antarctica (Spolaor et al., 2018). Both iodine and mercury in surface snow did
not show any diurnal cycle suggesting that a continuous recycling process may act on the snow
surface. Iodine shows an almost constant concentration in the first part of the experiment with some
oscillations, connected to sodium variations, hence possible sea spray deposition, occurring in the
second part of the experiment. While GEM still shows a clear diurnal cycle, the mercury in the
snow does not (Figure 2). The Hg concentration in the surface snow has some variations that are not
connected with changes in incoming solar radiation. As for the 2016 experiment, during the last
days of the experiment, a snowfall occurred, causing a rapid enrichment of iodine and Hg in the
surface snow followed by a rapid decrease most likely due to photo induced re-emission processes.

**5. Conclusions**
Three high temporal resolution experiments have been carried out between 2015 and 2017. The
three experiments were aimed at studying the behaviour of iodine and mercury (and bromine only
in 2016) in snow during the different polar seasons. One was conducted during the polar night (25th
to 29th of January 2017), one during the spring when the night and day cycle was present (6th to 10th
of April 2016) and one during late spring when sunlight was present for 24 hours a day (28th of
April to 1st of May 2015). The results obtained show that these elements have markedly different
behaviours in surface snow that are mainly governed by sunlight and snow deposition. For iodine,
the highest snow concentrations were detected during the winter polar night experiment (2017),
while the lowest were during late spring (2015) when continuous solar radiation reaches the snow
surface. For mercury the highest concentrations were detected in the winter (2017) and during late
spring experiment (2015).
Our high temporal resolution experiments did not have the aim of characterizing the average
surface snow concentrations but were designed to understand the behaviour of these elements in
surface snow within specific seasonal changes that can occur. A clear diurnal cycle for mercury and
iodine has been determined when a day and night cycle was still present, however, for Br (and its
enrichment factor) no diurnal cycle has been detected showing it has a more conservative behaviour
in snow. Total mercury concentrations in surface snow peak during the night and decreases during

the day, the opposite of its behaviour in the atmosphere. The daily variation in atmospheric GEM concentration might also be influence by changes in the boundary layer height, however the stable meteorological conditions during the experiment tended to minimize this effect. Iodine, acts similarly to mercury, peaking during the night and decreasing during the day. Considering our finding that up to 70% of the iodine present in the snow can be released to the atmosphere by photo-induced reactions. the active role of snow in providing gas phase iodine should be considered in studies of nucleation processes in the polar atmosphere.

This unique set of experiments has demonstrated for the first time the different behaviours of these target elements under different irradiation conditions and demonstrate that snow is an active substrate. The results obtained in Arctic snow could be translated to alpine regions and, more generally, anywhere in the presence of snow. The diurnal cycle determined for mercury in the Arctic, if demonstrated occurring in other places with high snow cover, could have an impact on water resources, with higher concentrations of mercury deposited in the water basin at night. These experiments have underlined some specific processes that can occur in surface snow, however additional studies are planned to better understand the real impact of these processes on the overlying atmosphere. We hope that these results contribute to the efforts in understanding the role of the snow pack in the Arctic mercury and iodine cycles and bromine behaviour in surface snow. Understanding the behaviour of these elements in the surface snow-pack may shed light on the role and the contribution of snow emissions, primarily to the marine boundary layer. For example, species such as iodine, are directly active in the formation of cloud condensation nuclei that could have a direct effect on polar climate.

**Author contribution**
A.S., E.B., D.C. conceived the experiment; A.S., E.B., D.C., F.G., F.D. collected the samples; A.S., C.T., F.L., E.B. measured the samples; M.M., M.Mat. provide the meteorological and radiation data; K.A.P. provide the mercury atmospheric data; A.S. ASL, WRLC, HA wrote the paper with inputs from A.D., C.B., M.B.

**Acknowledgements**
This project has received funding from the European Union's Horizon 2020 research and innovation programme under grant agreement No 689443 via project iCUPE (Integrative and Comprehensive Understanding on Polar Environments). We thank the support received at the Dirigibile Italia Arctic Station from the National Research Council — Department of Earth System Science and Environmental Technologies (CNR-DSSTTA). We acknowledge the help of ELGA LabWater in providing the PURELAB Pulse and PURELAB Flex which produced the ultrapure water used in these experiments.

**TABLES**

**Table 1.** Concentration of iodine and its enrichment in surface snow ($I_{snow}$, $I_{enr}$), surface snow mercury ($Hg_{snow}$), atmospheric mercury ($Hg_{atm}$) and surface snow sodium ($Na_{snow}$) during the different experiments. Concentrations and standard deviation (in brackets) are calculated for the entire dataset; when marked with (*) indicates that the concentration has been calculated without considering the snow fall events.

| | $I_{snow}$ (ngg$^{-1}$) | $Na_{snow}$ (ngg$^{-1}$) | $Hg_{snow}$ (ngg$^{-1}$) | $Hg_{atm}$ (ngm$^{-3}$) | $I_{enr}$ |
|---|---|---|---|---|---|
| **2015 (day)** | 0.147 (0.162) | 3442 (1180) | 0.010 (0.006) | 1.45(0.18) | 10.7 (25.5) |
| **2015*** | 0.090 (0.027) | 3502 (1030) | 0.009 (0.004) | 1.46(0.19) | 4.59 (1.43) |
| **2016 (day\night)** | 0.167 (0.076) | 2041 (777) | 0.007 (0.008) | 1.35 (0.13) | 25.7 (46.4) |
| **2016*** | 0.142 (0.057) | 2317 (498) | 0.007 (0.009) | 1.40 (0.08) | 10.2 (3.28) |
| **2017 (night)** | 0.382 (0.175) | 1518 (749) | 0.009 (0.006) | 1.26 (0.07) | 44.3 (11.2) |
| **2017*** | 0.433 (0.185) | 1786 (770) | 0.008 (0.004) | 1.26 (0.06) | 41.8 (8.40) |

**Table 2.** Correlation coefficient between Iodine and sodium, bromine and sodium (only 2016) and atmospheric and snow mercury. The correlation is calculated for the entire dataset. When the correlation is marked with "*", this indicates that the correlation has been calculated without considering the snow fall events. During the 2016 experiment the correlation between $Hg_{snow}$ *vs* $Hg_{atm}$* has been detrended to highlight the antiphase between $Hg_{atm}$ and $Hg_{snow}$. The plus and minus indicate if the association is positive or negative, which the values in parenthesis are the p-values.

| | I *vs* Na | I *vs* Na* | Br *vs* Na | Br *vs* Na* | $Hg_{snow}$ *vs* $Hg_{atm}$ | $Hg_{snow}$ *vs* $Hg_{atm}$* |
|---|---|---|---|---|---|---|
| **2015** | 0.24 (0.052)+ | 0.63 (<0.01)+ | NA | NA | 0.18 (0.13)+ | 0.36 (0.011)+ |
| **2016** | 0.21 (0.041)+ | 0.62 (<0.01)+ | 0.91 (<0.01)+ | 0.74 (<0.01)+ | 0.12 (<0.01)+ | 0.43 (<0.01)+[**] |
| **2017** | 0.90 (<0.01)+ | 0.89 (<0.01)+ | NA | NA | 0.22 (0.05)+ | 0.062 (0.63)+ |

[**]detrended 0.61 (0.056)-

**FIGURES**

**Figure 1.** Location of the experimental area in the proximity of Ny-Ålesund research village (black rectangular – right panel) and the site of experiments (grey rectangular – left panel) behind the "Gruvebadet" Aerosol Laboratory. Red triangle shows the GEM measurements site. Maps from toposvalbard.npolar.no.

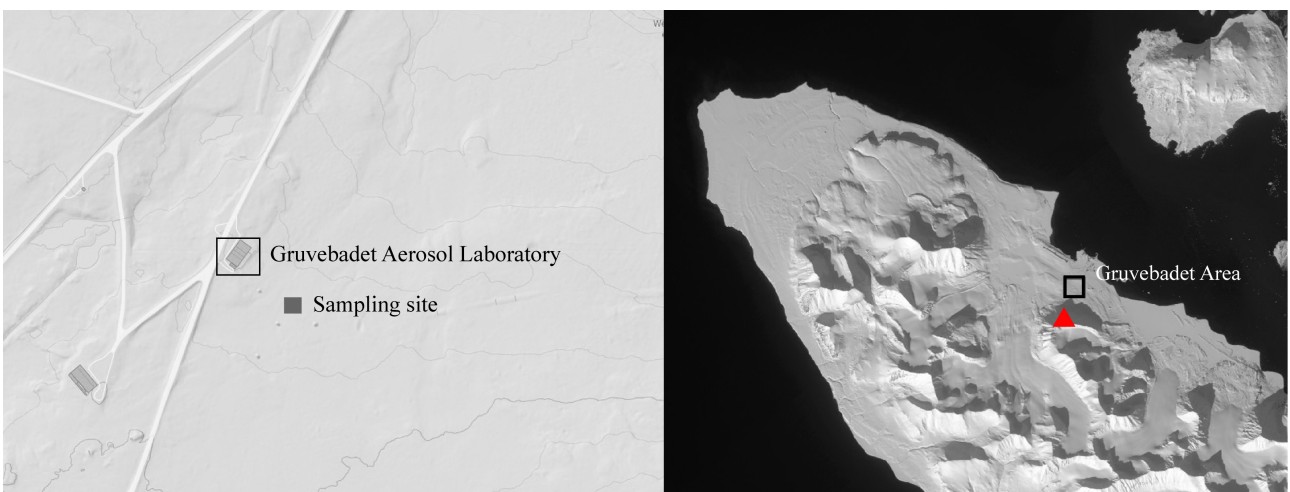

**Figure 2.** The 2015 experiment: continuous light conditions. The hourly sodium (g - dark red) concentrations are connected with iodine concentrations (f - light green for the raw data and green for the three-point smoothing) except during the snowfall where the signals decouple. Iodine enrichment (e - dark green) demonstrates the effect of snowfall on iodine concentration in surface snow. Gaseous elemental mercury (c - blue) exhibit a diurnal pattern while total mercury in surface snow (grey line and black line three-point smoothing) does not. Snowfall occurrence is highlighted by the pink rectangle. Snow and air temperature (d - dark blue and red) show the diurnal cycle connected with incoming solar radiation (ISR) (a - solid yellow). Wind speed is not shown since it was almost constant during the entire experiment. Dashed vertical lines indicate local midnight time.

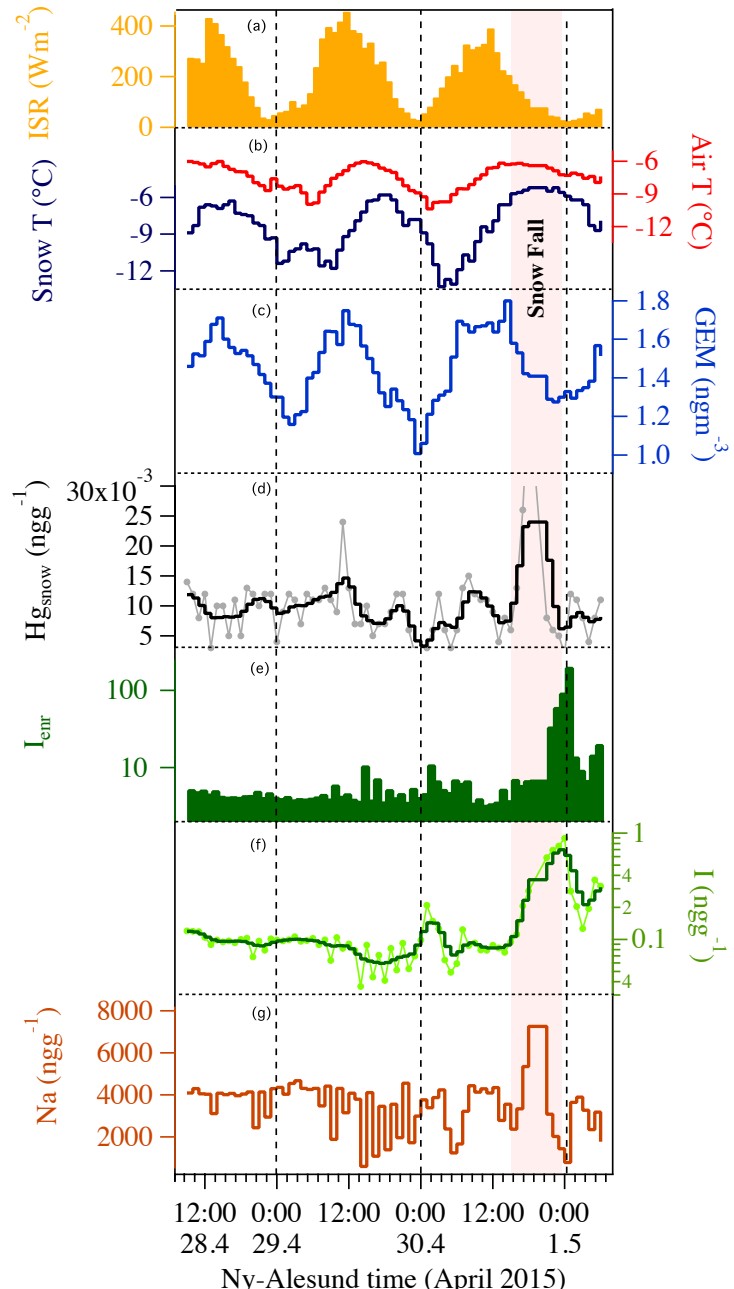

**Figure 3.** The 2016 experiment took place when a day and night cycle was available. Iodine
concentration (g - light green line for the raw data and green light for the three-point smoothing)
exhibited a diurnal variability (except during the snow fall event), not detected for sodium (h - dark
red line). The Iodine enrichment factor (f - dark green solid line) also exhibited a diurnal cycle and
highlights the effect of snowfall on iodine concentration in surface snow (pink rectangle shows the
snow fall event). De-trended GEM (d - blue line) and the surface snow de-trended total mercury
concentrations (grey lines for raw data and black line for the three-point smoothing) show opposing
diurnal patterns. Additional information can be found in Figure 4. Air temperature does not show a
pronounced diurnal cycle (b - red line) connected with incoming solar radiation (ISR)(a - yellow
solid). Wind speed is shown in grey (c). Dashed vertical lines indicate local midnight time.

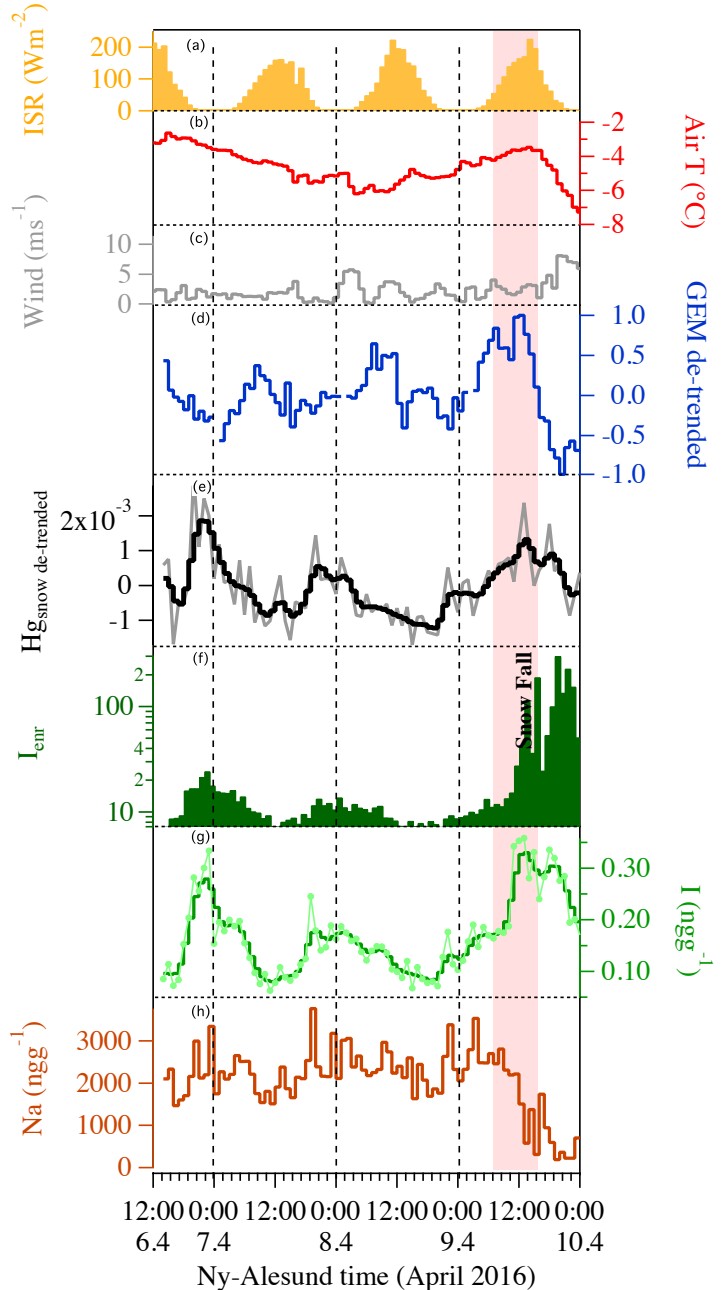


**Figure 4.** The lower panel shows the two series without any statistical treatment ($Hg_{atm}$=black;
$Hg_{snow}$=red). The regression line obtained for surface snow mercury is $Hg_{snow}$=-0.0004t + 16.136,
while for atmospheric mercury is GEM=-0.1127t + 4787.8. The middle panel shows the de-trended
Hg series in surface snow (in red/orange) and atmosphere (grey/black). The upper panel shows the
correlation between detrended $Hg_{snow}$ and $Hg_{atm}$ considering 6-hour average value. The figure is
based on the same data as Figure 3.

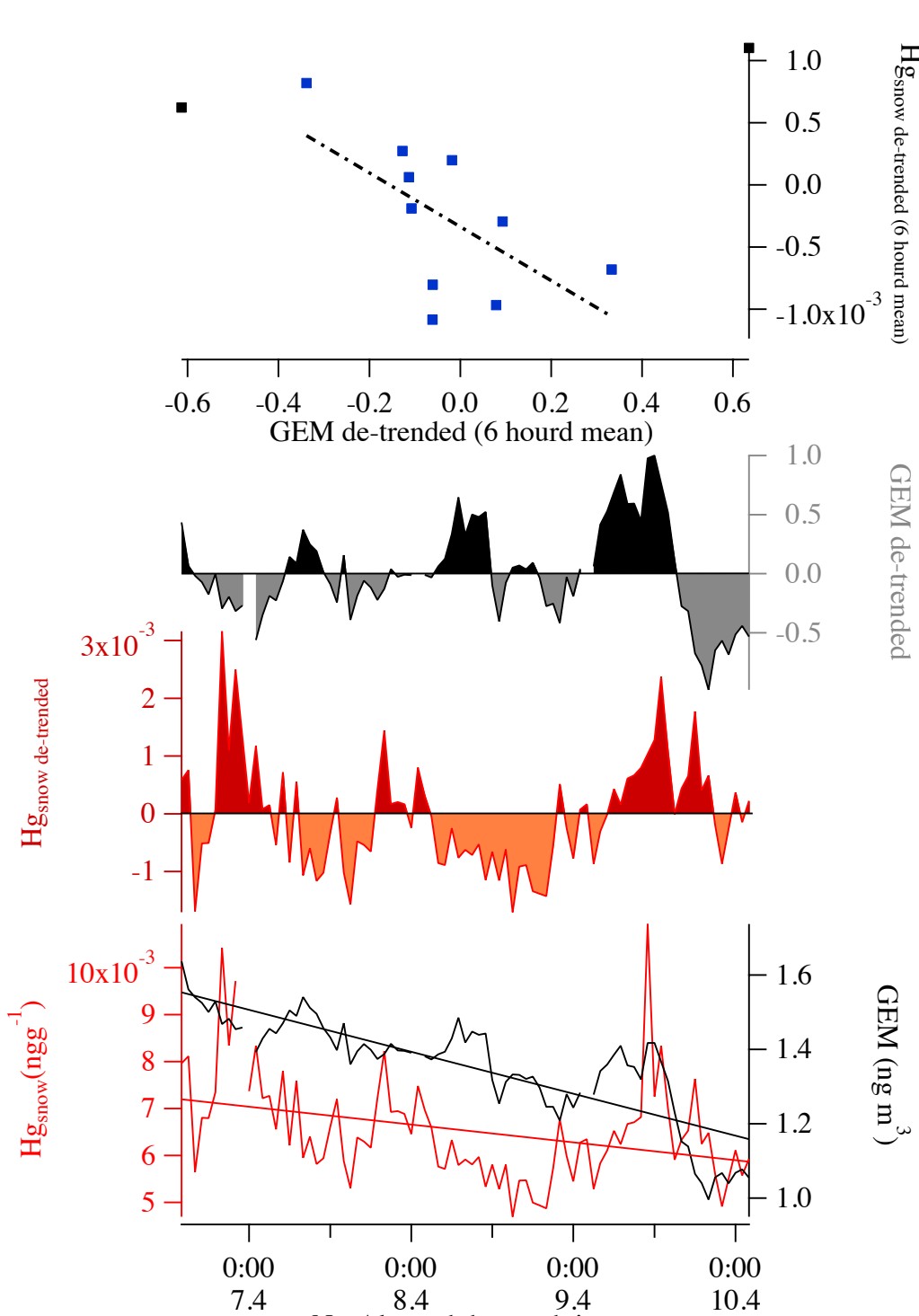


**Figure 5.** The 2017 experiment was conducted during the polar night. Iodine concentration (f -
green line) correlated with sodium concentration (g - dark red line). The Iodine enrichment factor (e
- dark green solid line) did not exhibit any diurnal cycle and had the higher value compare the three
experiments. Gaseous elemental mercury (c - blue line) and the surface snow total mercury
concentrations did not exhibit any diurnal pattern (d - light grey line for raw data and black line for
three-point smoothed). Snow and air temperature (b - dark blue and red) did not show any diurnal
cycle. Wind speed is shown in grey. Dashed vertical lines indicate local midnight time.

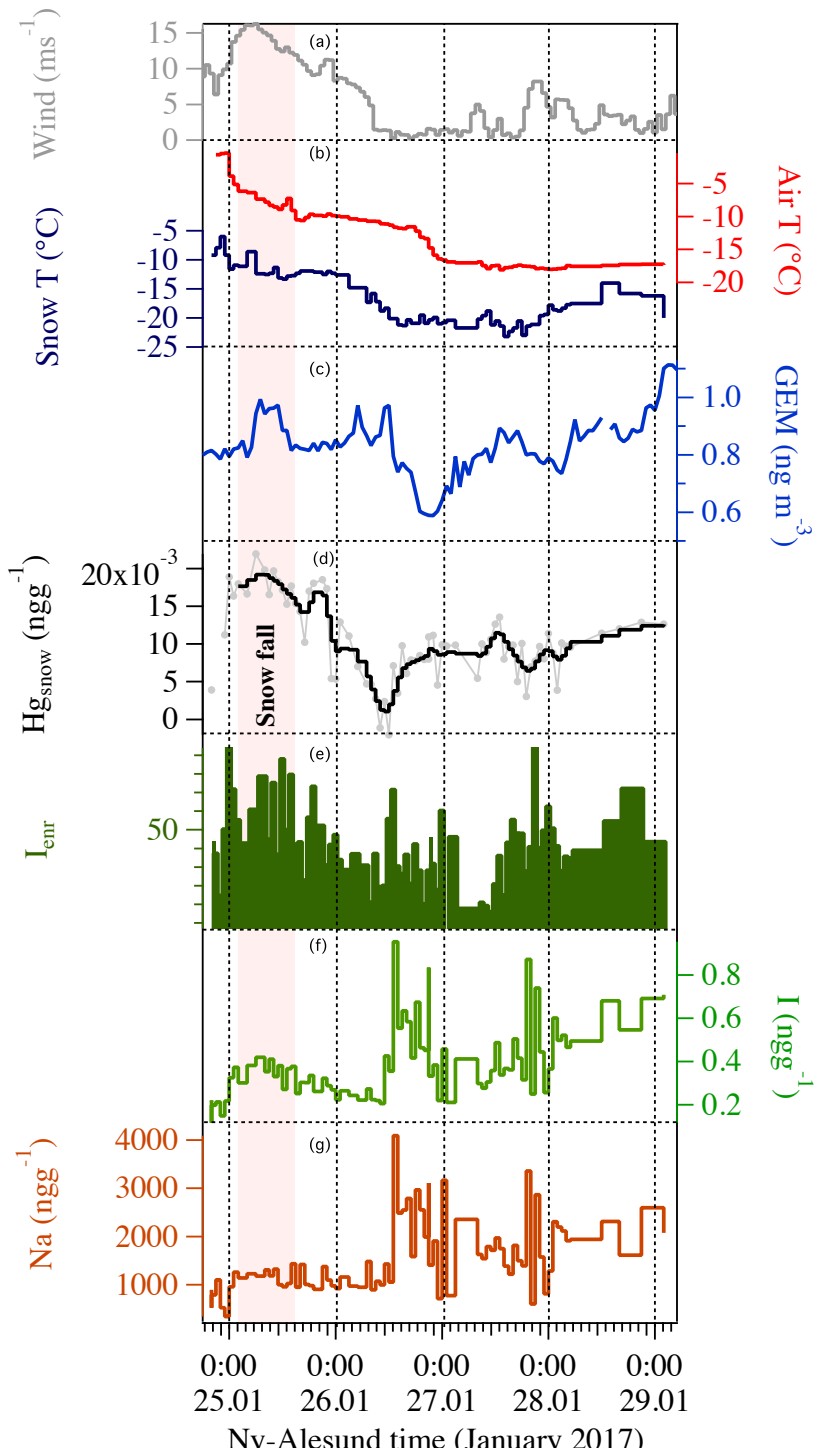


**Figure 6.** Surface Bromine recycle during the 2016 experiment. The Bromine concentration (light
blue line) does not show a diurnal variability and follows the sodium surface concentration (dark
red line). Bromine enrichment factor (blue solid line calculated as $Br_{enr}=Br_{snow}/(Na_{snow} \times 0.006)$
where 0.006 is the Br\Na sea water mass ratio) do not show a diurnal cycle but it is evident that
snowfall effects the bromine concentration and its enrichment factor during snowfall (pink
rectangle). Air temperatures do not show a pronounced diurnal cycle (red line) connect with the
incoming solar radiation (solid yellow). Dashed vertical lines indicate local midnight time.

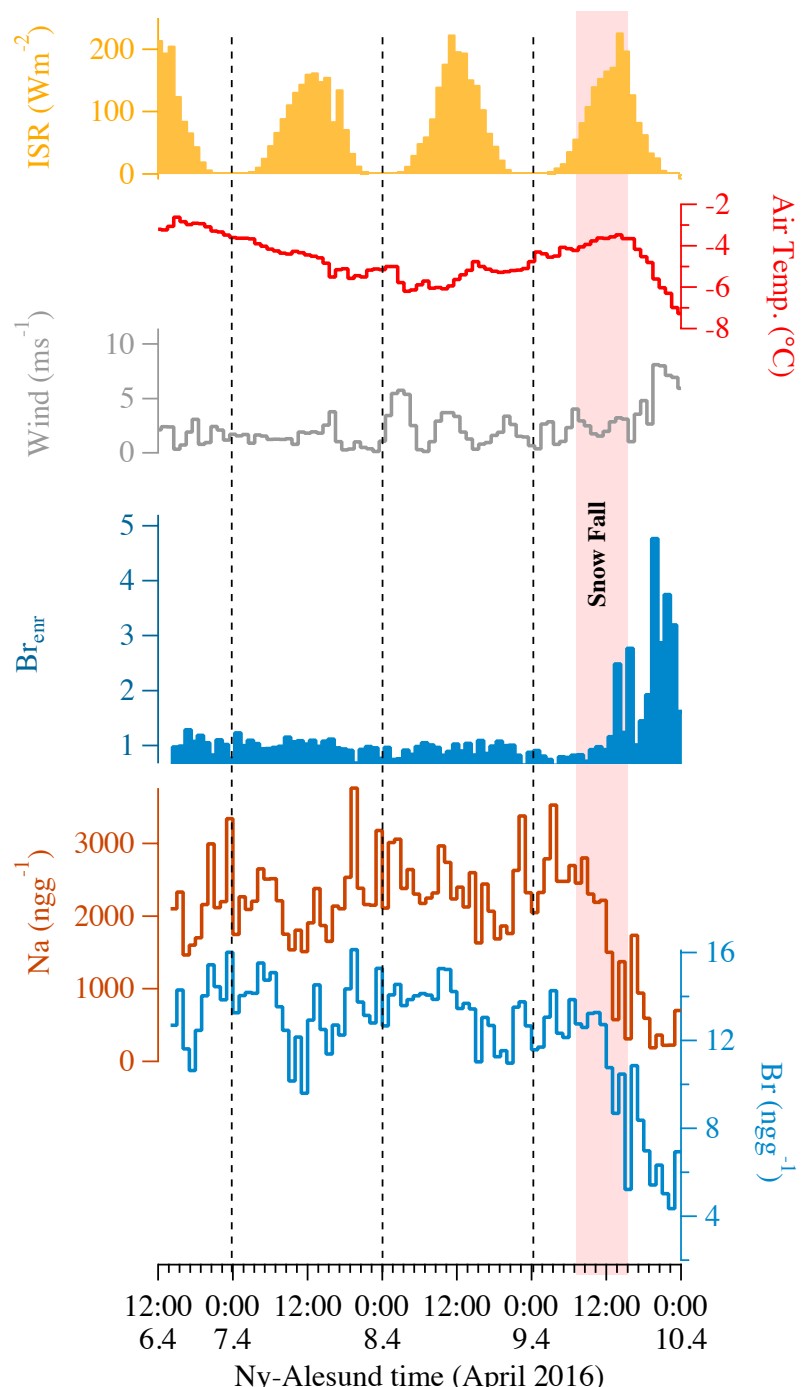

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
