# Peer review of "Diurnal cycle of iodine, bromine and mercury concentrations in Svalbard surface snow"

_Atmospheric Chemistry and Physics, 2019_

## Referee Comment (RC1) · Henrik Skov (Referee) · 17 Jul 2019

Review

This paper presents an important study of the behavior of mercury and iodine in surface snow. The study is carried out over 3 times 3 days with high time resolution. The study present some new and important results that clearly is of interest for ACP's readers. Therefore, the paper is recommended to be published after mayor corrections. General questions and recommendations The experimental design gives some short comings that need to be included in the interpretation of data: Representativeness Light is penetrating far below 3 cm depth depending on the snow morphology and the properties of the upper 3 cm can vary substantially. A discussion of how these factors affect the

results has to be included and used in the discussion of the results e.g. can there be a photo-reduction below or can there be a diffusion downwards, how is the "collapse" of snow pack affecting the concentrations.... The 2015 and 2016 experiments were made during spring where atmospheric Mercury depletion episodes (AMDE) are taken place potentially leading to high levels of GOM concentrations and thus followed by large GOM dry depositions (Skov et al. 2006[1]; Brooks et al. 2006[2]) leading to high Hg concentrations in surface snow, which then decreases over the coming days. The situation during the short measurement periods with stable GEM concentrations and relatively low snow concentrations indicates that AMDE does not occur. A discussion of the meteorological and chemical situation before the campaign has to be included As a consequence, it is difficult to judge to what degree the study really represent a seasonal pattern? This has to be further discussed in the paper, see also detailed comments.

Detailed comments Page 1 Line 1; Title: As Bromine is also treated and discussed, I suggest changing title to e.g. "Diurnal cycle of iodine, bromine and mercury concentrations in Svalbard surface snow

Page 3 line 79 and 80: in the list of papers I suggest to add Brooks et al. 2006[2] and Skov et al. 2006[1] as they represent to my knowledge the only study where the dry deposition of GOM and reemission of GEM has been determined. Line 85 to 87: the three campaigns are for sure representing different environmental conditions. They are also "only" snapshots and I miss a discussion of the period is typical for the season. The spring values are for sure outside AMDE episodes and thus was there any before the campaigns. The snow concentrations indicate that is not the case as AMDE has lead to high Hg concentrations in snow. The lack of AMDE has also to be used in the discussion of Br dynamics in snow. Page 4 line 115 Add Brooks et al. 2006 should be added. Line 117-118. Very high reemission rates have recently been measured Kamp et al. 2017. Similar situation might affect your results? Page 5 line 170; You have chosen to look on the upper 3 cm of the snow. The sun can penetrate much further

into to the snow pack and there can be diffusion between the upper layer and those below. A discussion how that might affect your study is needed.

Page 6. Line 195. You base your Hg analysis of the analysis on 202Hg other isotopes could have been very interesting but I guess that you have investigated the possibilities for those analyses? Line 206. You look on I and Br, it would have been interesting to see the variation on Cl as well. Page 7 Line 217. The location should be indicated as it is not at the snow sampling site and this is first told in the discussion section and I foresee some problem in the use of GEM data to explain the results as your snow sampling site. Page 8 line 258: You look on four elements therefore you need to change your title line 271 Your measurements are within the period of AMDE but clearly you are not affected directly by an episode as earlier noticed. Therefore, you need to discuss how your "snap shot" campaign fit into the seasonal behavior of GEM.

Page 9 line 312. The reactions of iodine species in high Arctic with NO3 will in general be low as NOx is very low <1 ppbv. It can be relevant close to Arctic cities. Further explanation is thus needed. Page 11 line 365. The deposition of Hg can be either dry or wet. It looks like the authors only consider wet deposition? The balance of dry deposition of GOM and remission of GEM has been determined by Brooks et al. 2006.

Line 382-391. Further discussion of the effect of reemission on the atmospheric GEM concentration needs to be elaborated. The reaction in the snow is not only taken place in the first 3 cm. The concentration of GEM will depend on the emission rate but also on the mixing height of the troposphere. The mixing height may vary even at a snow-covered site between night and day. Page 12. The description of Zeppelin station has to be moved to section 2.5 Line 411-414 I will argue that the Aspmo et al. 20053 paper indicates that very often the measurements at 400 m is not representative for surface GEM concentrations. This statement is even more correct considering the short campaign period. The text needs to be revised. Page 12 last line and page 13 line 428 Br has often been observed to be depleted from snow surfaces. These observations need to be discussed as this illustrates that there is discrepancies in

literature (Simpson et al. 20074). Line 449 In the conclusion indicate the exact period of the campaigns as they are fundamental in order to understand the conclusion Page 14. Line 464 to 465 Boundary layer height needs to be included here. Line 472 -473 The chemistry in the snow is not necessary comparable with alpine regions. Though at high altitudes, the NOx and VOC levels are most likely higher than in the Arctic and how will they affect the snow chemistry, further discussion is needed?

Page 17. Figure 2 and 3. The colors are very difficult to distinguish. I needed to have assistance in order to separate out the different variables. Redraw the figures and call them sub-figures a,b,c. . .

Page 19 Figure captions. Add; "The figure is based on the same data as Figure 3. . .."

List of suggested papers: 1. Skov, H.; Brooks, S.; Goodsite, M. E.; Lindberg, S. E.; Meyers, T. P.; Landis, M.; Larsen, M. R. B.; Jensen, B.; McConville, G.; Chung, K. H.; Christensen, J., The fluxes of Reactive Gaseous mercury measured with a newly developed method using relaxed eddy accumulation. Atmos. Environ. 2006, 40, 5452-5463. 2. Brooks, S.; Saiz-Lopez, A.; Skov, H.; Lindberg, S.; Plane, J. M. C.; Goodsite, M. E., The mass balance of mercury in the springtime polar environment. Geophys. Res. Lett. 2006, 33, L13812. 3. Aspmo, K.; Gauchard, P. A.; Steffen, A.; Temme, C.; Berg, T.; Bahlmann, E.; Banic, C.; Dommergue A; Ebinghaus, R.; Ferrari, C.; Pirrone, N.; Sprovieri, F.; Wibetoe, G., Measurements of atmospheric mercury species during an international study of mercury depletion events at Ny-Alesund, Svalbard, spring 2003. How reproducible are our present methods? Atmos. Environ. 2005, 39 (39), 7607-7619. 4. Simpson, W. R.; von Glasow, R.; Riedel, K.; Anderson, P.; Ariya, P.; Bottenheim, J.; Burrows, J.; Carpenter, L. J.; Friess, U.; Goodsite, M. E.; Heard, D.; Hutterli, M.; Jacobi, H. W.; Kaleschke, L.; Neff, B.; Plane, J.; Platt, U.; Richter, A.; Roscoe, H.; Sander, R.; Shepson, P.; Sodeau, J.; Steffen, A.; Wagner, T.; Wolff, E., Halogens and their role in polar boundary-layer ozone depletion. Atmos. Chem. Phys. 2007, 7 (16), 4375-4418.

---

## Referee Comment (RC2) · Anonymous Referee #1 · 2 Aug 2019

"Diurnal cycle of iodine and mercury concentrations in Svalbard surface snow," by Spolaor et al. conducted 3 high temporal resolution field campaigns between 2015 and 2017 at Ny-Alesund (Svalbard). Although previous studies show bidirectional exchange and indicate post-depositional processing of Hg, I, Na, and Br, little is known about the diurnal behavior of these species (especially iodine and mercury) and their interaction in surface snow. These experiments investigated the diurnal behavior of iodine, mercury, sodium, and bromine in the surface snowpack during varying polar seasons: 1) In 2017 during the polar night; 2) In 2016 during the spring when the night and day cycle are present; and 3) In 2015 during late spring with 24 hour sunlight. Governed mainly by sunlight and deposition, these elements have distinctly different behaviors in surface snow. These experiments show for the first time the varying behaviors of I, Hg, Br, and

Na in polar surface snow, thus reinforcing the fact that the polar snowpack is an active substrate for photochemical activity. They found that the highest iodine and mercury concentrations in surface snow occurred during the winter polar night while the lowest concentrations of iodine and mercury in surface snow occurred during the day/sunlight periods. The authors quantify that up to 70% of iodine present in snow can be released into the overlying atmosphere via photo-induced reactions, which has implications for polar boundary layer chemistry, climate, and particle formation. Bromine (and its enrichment factors) did not exhibit a diurnal cycle. All instruments made measurements of Hg, I, Br, and Na with RSD values < 5%. This study provides novel results in regard to the diurnal variability of Hg, I, Br, and Na in the polar snowpack surface as a function of select polar seasons. My only recommendation is that the authors consider providing more in-depth and specific content pertaining to the practical implications of their study (e.g., from the perspective of modeling, field studies, air and water quality, climate, etc) . It is an excellent manuscript, and I recommend it for publication.

---

## Author Comment (AC1) · 26 Sep 2019

Reply to Anonymous Referee #1

Diurnal cycle of iodine and mercury concentrations in Svalbard surface snow, by Spolaor et al. conducted 3 high temporal resolution field campaigns between 2015 and 2017 at Ny-Alesund (Svalbard). Although previous studies show bidirectional exchange and indicate post-depositional processing of Hg, I, Na, and Br, little is known about the diurnal behavior of these species (especially iodine and mercury) and their interaction in surface snow. These experiments investigated the diurnal behavior of iodine, mercury, sodium, and bromine in the surface snowpack during varying polar seasons: 1) In 2017 during the polar night; 2) In 2016 during the spring when the night and day cycle

are present; and 3) In 2015 during late spring with 24 hour sunlight. Governed mainly by sunlight and deposition, these elements have distinctly different behaviors in surface snow. These experiments show for the first time the varying behaviors of I, Hg, Br, and Na in polar surface snow, thus reinforcing the fact that the polar snowpack is an active substrate for photochemical activity. They found that the highest iodine and mercury concentrations in surface snow occurred during the winter polar night while the lowest concentrations of iodine and mercury in surface snow occurred during the day/sunlight periods. The authors quantify that up to 70% of iodine present in snow can be released into the overlying atmosphere via photo-induced reactions, which has implications for polar boundary layer chemistry, climate, and particle formation. Bromine (and its enrichment factors) did not exhibit a diurnal cycle. All instruments made measurements of Hg, I, Br, and Na with RSD values < 5%. This study provides novel results in regard to the diurnal variability of Hg, I, Br, and Na in the polar snowpack surface as a function of select polar seasons. My only recommendation is that the authors consider providing more in-depth and specific content pertaining to the practical implications of their study (e.g., from the perspective of modeling, field studies, air and water quality, climate, etc). It is an excellent manuscript, and I recommend it for publication.

Reply: We thank the referee for their positive evaluation. As suggested, we added to the end of the conclusions section possible practical implication for this study. The text have been improved at line 556-561

Text: We hope that these results contribute to the efforts in understanding the role of the snow pack in the Arctic mercury and iodine cycles and bromine behaviour in surface snow. Understanding the behaviour of these elements in the surface snowpack may shed light on the role and the contribution of snow emissions, primarily to the marine boundary layer. For example, species such as iodine, are directly active in the formation of cloud condensation nuclei that could have a direct effect on polar climate.

---

## Author Comment (AC2) · 26 Sep 2019

Review This paper presents an important study of the behaviour of mercury and iodine in surface snow. The study is carried out over 3 times 3 days with high time resolution. The study present some new and important results that clearly is of interest for ACP's readers. Therefore, the paper is recommended to be published after mayor corrections. General questions and recommendations. The experimental design gives some short comings that need to be included in the interpretation of data: Representativeness Light is penetrating far below 3 cm depth depending on the snow morphology and the properties of the upper 3 cm can vary substantially. A discussion of how these factors affect the results has to be included and used in the discussion of the results e.g. can there be a photo-reduction below or can there be a diffusion downwards, how is the

"collapse" of snow pack affecting the concentrations: The 2015 and 2016 experiments were made during spring where atmospheric Mercury depletion episodes (AMDE) are taken place potentially leading to high levels of GOM concentrations and thus followed by large GOM dry depositions (Skov et al. 20061; Brooks et al. 20062) leading to high Hg concentrations in surface snow, which then decreases over the coming days. The situation during the short measurement periods with stable GEM concentrations and relatively low snow concentrations indicates that AMDE does not occur. A discussion of the meteorological and chemical situation before the campaign has to be included As a consequence, it is difficult to judge to what degree the study really represent a seasonal pattern? This has to be further discussed in the paper, see also detailed comments.

Reply: We thank the referee for all their comments and suggestions. The text has been improved and all the detailed comments have been addressed and the text modified accordingly. Specific replies to general comments and recommendations are presented below. In red the text add directly into the manuscript

Detailed comments

Page 1 Line 1; Title: As Bromine is also treated and discussed, I suggest changing title to e.g. "Diurnal cycle of iodine, bromine and mercury concentrations in Svalbard surface snow"

Reply: We thank the referee for the suggestion and the title has been changed accordingly.

Text: Diurnal cycle of iodine, bromine and mercury concentrations in Svalbard surface snow

Page 3 line 79 and 80: in the list of papers I suggest to add Brooks et al. 2006 and Skov et al. 2006 as they represent to my knowledge the only study where the dry deposition of GOM and reemission of GEM has been determined.

[Figure]

Reply: We have added the suggested references

Line 85 to 87: the three campaigns are for sure representing different environmental conditions. They are also "only" snapshots and I miss a discussion of the period is typical for the season.

Reply: Results discussed here are only representative of the experimental period and should not be considered as a reference for a specific season but give some general indication. The sampling periods were mainly selected for their different light conditions, since the experiment was mainly designed to catch the effects of incoming radiation during a representative part of the season. During the winter period, we chose the second half of January, because in December snow cover is not homogeneously distributed and is incomplete (López-Moreno et al 2016). Windblown dust from ground left exposed could affect the snow surface concentrations. The spring periods were chosen for the clear day/night cycle, we wanted to have a complete solar irradiation cycle and we hoped to capture possible halogen and mercury deposition events connected with bromine explosion events in the north and concurrent AMDEs. However, we were unlucky, as the eastern and northern coast of Svalbard was devoid of sea ice during the sampling period making AMDEs less likely (in addition Kongsfjorden is not covered by sea ice since 2011). The third experiment was scheduled at the beginning of May to have a full 24h of sunlight reaching the snow surface while the temperature remained below freezing to avoid additional complications caused by snow pack melt or collapse. Meteorological situation prior the campaigns periods where evaluate with particular focusing on temperature and wind speed. A figure (Figure 1) showing the meteorological condition has been include to this specific reply. The temperature (left panels in the figure 1) shows the long-term daily mean temperature for the respective campaign month (black dots), $\pm$ standard deviation (grey bars). In this case "long-term" refers to the recent 20-year period, with the respective year excluded for the calculation of the mean. The orange line marks the daily mean temperature of the respective campaign month, Red colour indicates the actual campaign days at the end

of this period. From these plots is possible determining the representativeness of the select periods in term of meteorological condition. The right panels follow the same representation, but for wind. Daily mean wind speed is not a realistic value, but it gives and indication for generally stormy days with likely increased snow drift. We prefer to do not include these information directly in the text since is not the aim of the paper however we improve the text at lines 170-184 describing the strategy adopted for the experiment period selection.

López-Moreno, J. I., J. Boike, A. Sanchez-Lorenzo and J. W. Pomeroy (2016). "Impact of climate warming on snow processes in Ny-Ålesund, a polar maritime site at Svalbard." Global and Planetary Change 146: 10-21.

Text: The 2017 experiment was conducted during the second half of January when full snow cover is guaranteed (López-Moreno et al., 2016). In December, snow cover in the Spitsbergen area is not homogeneously distributed. The ground could still be partially exposed, meaning that locally generated windblown dust could affect the trace element concentrations in the snow surface. The spring period selected for the 2016 experiment had two main characteristics: a well-defined night and day cycle without a long sunset, avoiding possible incoming solar radiation by diffraction processes over the horizon. There was also the possibility to observe atmospheric mercury depletion events (AMDE) connected with bromine explosion events (Lu et al., 2001; Moore et al., 2014; Schroeder and Munthe, 1998). Unfortunately, these events were not observed as the northern coast of Svalbard was virtually ice free by the time we started sampling. The 2015 experiment was scheduled to end at the beginning of May, when we have a full 24 h of sunlight reaching the snow surface, but temperatures are still below freezing, avoiding/minimizing the confounding effects of snow pack melt or collapse on surface photochemical processes and gaseous mercury transport in the interstitial air. The meteorological conditions throughout all the experiments are within the expected local conditions for the time of year".

The spring values are for sure outside AMDE episodes and thus was there any before

the campaigns. The snow concentrations indicate that is not the case as AMDE has lead to high Hg concentrations in snow. The lack of AMDE has also to be used in the discussion of Br dynamics in snow.

Reply: The experiment was not designed to specifically follow an AMDE, but the effect of sunlight/photo-chemical reactions on the snow surface concentrations of Iodine, Bromine and mercury. Considering the paper of Angot et al. 2106, AMDEs occured in 15% of the 2011-2014 springtime GEM observations at Zeppelin. Sea-ice dynamics across the Arctic might influence the interannual variability of AMDEs; AMDEs occured at Zeppelin in 18% (2011), 13% (2012), 16% (2013), 20% (2014), and only 6% (2015) of the springtime observations. The Angot et al 2016 review on AMDEs reports that such events are more rare at Zeppelin than at other Arctic sites; AMDEs occurred in 39%, 28%, and 15% or the 2011-2014 springtime observations at Alert, Station Nord, and Zeppelin, respectively. If an AMDE occurred, results would have been very interesting, however it was unlikely to occur in the time period of our experiments. Some AMDE events may have occurred before the experiment but this was not the main aim of the experiment. We agree with the referee, that some more information about the possibility of AMDE events occurring during the experiment should have been added. However, to catch an AMDE event, the experimental design, and the time period of the year would have had to have been modified to get a longer time coverage with a lower temporal resolution, with a period when sea ice breakup was more likely. The text has been improved and a discussion regarding possible AMDE has been included. The text has been added to the discussion section at lines 344- 352

Text: Atmospheric mercury depletion events (AMDE) can occur during the springtime causing large-scale deposition of mercury to the snow pack concurrently with ozone photochemistry and oxidation reactions involving bromine. During our spring experiments we have not observed any rapid decreases in GEM or increases in mercury concentrations in the surface snow. This indicates that no AMDE occurred during the sampling periods and that, especially for bromine, the main depositional source was

from sea spray given the distance from the coast line (< 1km) and the positive correlation with Na (Table 2). This is inline with the findings of (Angot et al., 2016a), who reported that AMDEs occur much less frequently at Zeppelin station than they do at Alert or Station North in Greenland".

Page 4 line 115 Add Brooks et al. 2006 should be added.

Reply: We have added the reference as suggested

Line 117-118. Very high reemission rates have recently been measured Kamp et al. 2017. Similar situation might affect your results?

Reply: The conditions at station North and Ny-Alesund are very different mainly due to the absence of sea ice close to the Svalbard coast and the bromine chemistry that is associated with it. The absence of Bromine chemistry connected with sea ice is also reflected in the Br enrichment value which is close to the sea water ratio, no additional Br sources were found during the experiment. In addition, during the spring experiments the wind speed was around 3 ms-1 suggesting different conditions to those during the Kamp et al. 2017 experiment. While it is more difficult to estimate an emission flux during the day and night oscillation for the 2016 experiment, we can estimate the mercury released after snow deposition occurred. The emission rate is calculated by considering the sampling depth of 3 cm for an area of 1 m2 and considering an average snow density of 0.3 g cm-3. The mercury snow concentration decreased from 45 to 8 pgg-1 with a net loss of 37 pgg-1 of total mercury in 1 hour. Assuming all snow mercury lost is lost as GEM, the emission rate would be 5.5 ng m-2 h-1, a similar order of magnitude to that determined by Kamp et al. 2017. However, Kamp et al. 2017 measured the total emission flux while we focussed on the upper snow pack layer, emissions from the lower/deeper strata are not considered. The text has been improved at line 123 and at line 470-477.

Text: High gaseous elemental mercury (GEM) emission from the snow pack has also been determined at Station North (Greenland) where the emission flux can rise up to

190 ng m-2 min-1 (Kamp et al., 2018)".

Text: After the snow fall the mercury surface snow concentration decreased from 45 to 8 pgg-1 with a net loss of 37 pgg-1 of total mercury in 1 hour. Assuming all snow mercury lost is lost as GEM, considering a sampling depth of 3 cm for an area of 1 m2 and considering an average snow density of 0.3 g cm-3, the emission rate would be 5.5 ng m-2 h-1, a similar order of magnitude to that determined by Kamp et al. 2017. It must be noted that Kamp et al. 2017 measured the total emission flux while we focussed on the upper snow pack layer, emissions from the lower/deeper strata are not considered that might contribute to the total emission from the snow pack".

Page 5 line 170; You have chosen to look on the upper 3 cm of the snow. The sun can penetrate much further into to the snow pack and there can be diffusion between the upper layer and those below. A discussion how that might affect your study is needed.

Reply: We agree with the referee and the text has been modified to reflect this comment. It is true that light can penetrate deeper into the snow pack and affect/promote photochemical reactions in the deeper snow layers. We chose to sample only the upper 3 cm to avoid sampling different snow layers, with different snow crystal shapes and size that might have a significantly different density. Focusing on the surface layer minimized possible effects of these physical parameters on the data interpretation. The upper 3 cm could be affected by the diffusion of gaseous iodine and mercury from the lower layers released by photoactivation. Previous studies (including the papers highlighted by the referee) suggest that mercury and iodine re-emission occurs quite quickly. However, mercury released from lower levels should not affect snow concentrations in the upper 3 cm as it exists as GEM within the pores of the snowpack and is generated from the reduction of Hg2+ (Fain et al 2007). We directly measured the total Hg in the snow by ICP-SFMS. The upper 3 cm are also particularly sensitive to night-time mercury and iodine deposition. Increasing the sampling depth would result in "smoothing" the night time deposition signal. It is not impossible for day time re-emission from the snow to reach the deeper layers, but during the polar-night it is

much unlikely that mercury in deposited snow can diffuse and dissolve into the deeper snow layers. The text has been modified between lines 193-197.

Text: This choice also minimizes the effect of different physical snow conditions (density and crystal shape and size). Although re-emission of mercury and iodine from lower snow strata could influence the gaseous concentrations in the snow interstitial air (Faïn et al., 2007) it is much less likely to have a direct effect on snow concentrations due to its poor solubility in water".

Page 6. Line 195. You base your Hg analysis of the analysis on 202Hg other isotopes could have been very interesting but I guess that you have investigated the possibilities for those analyses?

Reply: Unfortunately, our lab is not equipped with a multi detector ICP-SFMS that is needed to be able to measure mercury stable isotope ratios in the samples. We tried using our instrument, but the measurement uncertainties were an order of magnitude higher than the natural variability caused by fractionation. You need a truly simultaneous instrument to measure the isotope ratios with enough precision. Another problem is that the faraday cups used in these instruments are less sensitive than the electron multipliers typically used in mono detector scanning instruments used for total metals analysis. The m/z 202 corresponds to the most abundant isotope of mercury and is free from potential interferences from tungsten or lead, an important consideration since we are measuring in low resolution mode to increase ion transport.

Line 206. You look on I and Br, it would have been interesting to see the variation on Cl as well.

Reply: Unfortunately chlorine has not been measured. Mercury, Bromine, Sodium and Iodine have been measured by ICP-SFMS. Chloride cannot be measure using ICP-MS in low resolution mode due to the interferences with the oxygen. We now have dedicated ion chromatographs and an ICP-AES instrument that may be able to detect Cl in snow in future studies.

Page 7 Line 217. The location should be indicated as it is not at the snow sampling site and this is first told in the discussion section and I foresee some problem in the use of GEM data to explain the results as your snow sampling site.

Reply: The text to reflect this has been added at line 243-244 and lines 244 - 249. The GEM measurements at the Zeppelin station were the only measurements with which we can compare our snow surface measurements. GEM measurements at the snow sampling site would have been much more reliable. However, we assume that the snow reactions/emission occurring in the snow at the sampling site, also occur in the snow surrounding Zeppelin station. The snow concentration might be slightly different but the general process such as the diurnal cycle should be similar. The experiment was designed and focused on determining changes in concentration not a characterization of the absolute concentration and snow pack mercury mass balance during the sampling period.

Text: Atmospheric mercury concentrations were obtained from the Zeppelin Observatory located at 474 m a.s.l, less than 1 km away from the sampling site (Figure 1)."

Text: The measurements at Zeppelin were the only GEM measurements available in the Ny-Alesund area. Although GEM measurements at the snow sampling site would have been more reliable in determining possible interactions between snow and atmospheric mercury, it was not possible to set up an instrument at the site. We assume that the snow reactions occurring at the sampling site at 40 m a.s.l. are of the same order of magnitude as those occurring in the snow layers surrounding Zeppelin station.

Page 8 line 258: You look on four elements therefore you need to change your title

Reply: We are essentially looking at three elements, sodium is only used for normalizing the Br and I from sea spray deposition that might affect their concentrations, it is a reference element, much like an internal standard in elemental analyses, as it is not photochemically reactive.

[Figure]

line 271 Your measurements are within the period of AMDE but clearly you are not affected directly by an episode as earlier noticed. Therefore, you need to discuss how your "snap shot" campaign fit into the seasonal behavior of GEM.

Reply: We have improved the text between lines 308 - 316. We have already explained that the experiment was designed to mainly determine the effect of sunlight on snow surface chemistry and not the general behaviour of mercury during a specific season. The experimental periods were chosen to avoid snow fall deposition in the middle of the experiment or periods with strong wind, and windblown snow transport (for this reason, the winter experiment was extended to 5 days) to minimize possible confounding meteorological effects and make the experiments more comparable. We cannot exclude that the same behaviour could have occurred over the entire season (obviously an AMDE would be a change in behaviour) or when the meteorological parameters (including snow cover, irradiance, wind strength and cloud coverage) were similar. In other periods of the year the cyclicity could be weakened and might not by visible. We have already specified in line 304 that the GEM concentration are only representative of the experimental period and should not be considered reference concentrations for a specific season, such a discussion is outside the scope of this manuscript and is the intellectual property of the operators of the mercury measurements at Ny Alesund, whom we thank for supplying the data for our observation period. Text: The experimental periods were chosen to reduce the possibility of snowfall deposition during the experiment and to avoid periods with strong wind and subsequent windblown snow transport (the main reason why the winter experiment was lengthened to 5 days). This was all done to minimize the effects of meteorological parameters on our results and make the experiments more comparable. We cannot exclude that the behaviour that we found for iodine, mercury and bromine could be significantly different during the specific season/periods (such as for example during an AMDE) or when meteorological conditions such as snow deposition frequency and amount, wind strength and cloud coverage were different".

Page 9 line 312. The reactions of iodine species in high Arctic with NO3 will in general be low as NOx is very low <1 ppbv. It can be relevant close to Arctic cities. Further explanation is thus needed.

Reply: We agree with the referee aand have added the following phrase "Although typical NOx levels are low in the Arctic, the reaction with NO3 could be relevant close to Arctic cities and under episodes of anthropogenic long range transport of pollution to the Arctic."The text have been modify at line 367-369

Text: Although typical NOx levels are low in the Arctic, the reaction with NO3 could be relevant close to Arctic cities and under episodes of anthropogenic long range transport of pollution to the Arctic."

Page 11 line 365. The deposition of Hg can be either dry or wet. It looks like the authors only consider wet deposition? The balance of dry deposition of GOM and remission of GEM has been determined by Brooks et al. 2006.

Reply: We have added to lines 422-424 to underline this. So, we obviously agree with the referee that mercury deposition can be either wet or dry or both. In the absence of snow fall, dry deposition should be the main driver of increases in surface snow concentrations while during the snow fall events, wet deposition will be the main driver of Hg fluxes to the snow pack.

Text: Atmospheric mercury can undergo wet or dry deposition to the snow pack, either as gaseous elemental (GEM) or oxidised mercury (GOM), and can be reemitted as GEM (Brooks et al. 2006)."

Line 382-391. Further discussion of the effect of reemission on the atmospheric GEM concentration needs to be elaborated. The reaction in the snow is not only taken place in the first 3 cm. The concentration of GEM will depend on the emission rate but also on the mixing height of the troposphere. The mixing height may vary even at a snow-covered site between night and day.

[Figure]

Reply: We agree that the mixing height of the troposphere can change, affecting the GEM concentration. The link between snow and atmospheric mercury has found several times with a dedicated field experiment at Dome C. The conditions at Dome C (Antarctic plateau) are quite different to Svalbard, especially since the mixing height can be very close to the surface (30 to 100 m height). In Svalbard the situation is more complex mainly due to the orography. The text has been improved at line 439 – 446 to explain the possible effects of mixing height on our experiments.

Text: This is not the only mechanism that can lead to increases in atmospheric concentrations. Changes in the atmospheric mixing layer height may lead to apparent concentration changes of atmospheric species, even if total amounts in the boundary layer remain constant. In the Ny-Ålesund area it is difficult to estimate the height of the boundary layer due to effects induced by winds and by the orography of the Brøgger Peninsula. However, during the experiments the stable meteorological conditions suggested that the atmospheric mixing height was quite stable, minimizing any influence of the boundary layer on GEM concentrations"

Page 12. The description of Zeppelin station has to be moved to section 2.5

Reply: We have modified accordingly

Line 411-414 I will argue that the Aspmo et al. 2005 paper indicates that very often the measurements at 400 m is not representative for surface GEM concentrations. This statement is even more correct considering the short campaign period. The text needs to be revised.

Reply: The text has been modified between lines 482 - 487. We assume that the snow reactions in the snow at the sampling site are also occurring in the snow surrounding Zeppelin station at similar rates. The snow concentrations may be slightly different but the general processes such as the diurnal cycle should be similar. So although we are not detecting the mercury actually emitted from our snow field, the order of magnitude and the trends should be comparable.

Text: Although the two sites may not be directly connected (Aspmo et al., 2005), we assume that the snow mercury and iodine release mechanisms that occur in the snow at our sampling site are also occurring in the snow surrounding the Zeppelin station at more or less the same rates. Consequently, GEM atmospheric concentrations and the diurnal cycle should be representative of the variations in the atmospheric cycle above the surrounding sampled snow field."

Page 12 last line and page 13 line 428 Br has often been observed to be depleted from snow surfaces. These observations need to be discussed as this illustrates that there is discrepancies in literature (Simpson et al. 20074).

Reply: Bromine can be depleted in some area as suggested by Simpson. However, the depletion should occur during atmospheric transport in the gas or aerosol phase, not after deposition. At Svalbard, at sea level, no post depositional processes have been observed. Other preliminary results suggest that in Antarctica, bromine after deposition is preserved. The complexity in bromine chemistry Br is mainly during transport and deposition processes. Wren et al Atmos. Chem. Phys., 13, 9789–9800, 2013, show that $Br_2$ can be released from saline aged acidic snow in the presence of $O_3$ at temperatures above the NaCl-water eutectic temperature. These conditions were not present during our experiment. Text has been improved at line 498 to show we considered this process and the text that follows shows it did not occur during our experiment.

Text: "recycling re-emission processes as suggest by previous studies (Simpson et al., 2007)"

Line 449 In the conclusion indicate the exact period of the campaigns as they are fundamental in order to understand the conclusion

Reply: The text has been modify accordingly

Pag 14. Line 464 to 465 Boundary layer height needs to be included here.

Reply: As has been previously reported, an estimate of the boundary layer height is

rather difficult in the Ny Alesund area. We have modified the text (line 541-543) to be more clear regarding possible boundary layer effects.

Text: The daily variation in atmospheric GEM concentration might also be influence by changes in the boundary layer height, however the stable meteorological conditions during the experiment tended to minimize this effect".

Line 472 -473 The chemistry in the snow is not necessary comparable with alpine regions. Though at high altitudes, the NOx and VOC levels are most likely higher than in the Arctic and how will they affect the snow chemistry, further discussion is needed?

Reply: This is only a possible hypothesis and a suggestion for further experiments. The behaviour of iodine and mercury can be completely different, but in light of the higher accumulation and different meteorological and climatic conditions in the Alps, doing similar experiments there would be interesting. We do not want to speculate but we would like to determine if these processes can also be detected in other snow-covered areas at lower latitudes.

Page 17. Figure 2 and 3. The colors are very difficult to distinguish. I needed to have assistance in order to separate out the different variables. Redraw the figures and call them sub-figures a,b,c

Reply:We modified them accordingly

Page 19 Figure captions. Add; "The figure is based on the same data as Figure 3"

Reply:We added the text suggested
* * *
[Figure]

**Fig. 1.**